# Liquid Chromatography-Tandem Mass Spectrometry Method Development and Validation for the Determination of a New Mitochondrial Antioxidant in Mouse Liver and Cerebellum, Employing Advanced Chemometrics

**DOI:** 10.3390/molecules30091900

**Published:** 2025-04-24

**Authors:** Anthi Panara, Dimitra Biliraki, Markus Nussbaumer, Michaela D. Filiou, Nikolaos S. Thomaidis, Ioannis K. Kostakis, Evagelos Gikas

**Affiliations:** 1Laboratory of Analytical Chemistry, Department of Chemistry, National and Kapodistrian University of Athens, Panepistimiopolis Zografou, 15771 Athens, Greece; ntho@chem.uoa.gr (N.S.T.); vgikas@chem.uoa.gr (E.G.); 2Laboratory of Pharmaceutical Chemistry, Department of Pharmacy, National and Kapodistrian University of Athens, Panepistimiopolis Zografou, 15771 Athens, Greece; dimitrab@pharm.uoa.gr (D.B.); ikkostakis@pharm.uoa.gr (I.K.K.); 3Laboratory of Biochemistry, Department of Biological Applications and Technology, University of Ioannina, 45110 Ioannina, Greece; nussbaumermarkus.80@gmail.com (M.N.); mfiliou@uoi.gr (M.D.F.); 4Biomedical Research Institute, Foundation for Research and Technology-Hellas, 45100 Ioannina, Greece

**Keywords:** triphenylphospine-hydroxytyrosol (TPP-HT), mitochondria, LC-MS/MS, cerebellum, liver, chemometrics, anxiety

## Abstract

Anxiety and stress-related disorders affect all ages in all geographical areas. As high anxiety and chronic stress result in the modulation of mitochondrial pathways, intensive research is being carried out on pharmaceutical interventions that alleviate pertinent symptomatology. Therefore, innovative approaches being currently pursued include substances that target mitochondria bearing an antioxidant moiety. In this study, a newly synthesized antioxidant consisting of triphenylphosphine (TPP), a six-carbon alkyl spacer, and hydroxytyrosol (HT) was administered orally to mice via drinking water. Cerebellum and liver samples were collected and analyzed using ultra-high-performance liquid chromatography-tandem triple quadrupole mass spectrometry (UHPLC-MS/MS) to assess the levels of TPP-HT in the respective tissues to evaluate in vivo administration efficacy. Sample preparation included extraction with appropriate solvents and a preconcentration step to achieve the required sensitivity. Both methods were validated in terms of selectivity, linearity, accuracy, and limits of detection and quantification. Additionally, a workflow for evaluating and statistically summarizing multiple fortified calibration curves was devised. TPP-HT penetrates the blood–brain barrier (BBB), with a level of 11.5 ng g^−1^ quantified in the cerebellum, whereas a level of 4.8 ng g^−1^ was detected in the liver, highlighting the plausibility of orally administering TPP-HT to achieve mitochondrial targeting.

## 1. Introduction

“Mental disorders are among the top 10 leading causes of health loss worldwide, with anxiety disorders and depression ranked as the most common across all age groups and locations”, as stated by the Institute of Health Metrics and Evaluation [1]. Ιn 2019, a population of 301 million was affected by stress-related disorders [2]. It is noteworthy that the prevalence of anxiety disorders in the post-COVID era has also significantly increased, highlighting the need for identifying effective therapeutic approaches for pertinent pathologies [3].

Anxiety disorders affect millions of individuals drastically, impacting their life quality, with the most frequently observed symptoms being insomnia, nausea, and the sense of tiredness and tension, as well as diminished social skills and the continuous fear of danger [4]. Psychological stress is a risk factor for all chronic conditions. Detrimental stress effects in brain function are well established [5], with stress exposure leading to increased anxiety levels and anxiety sensitivity [6,7]. If not addressed properly and timely, high anxiety can potentially lead to pathological conditions (e.g., panic disorder, social anxiety disorder, agoraphobia). Continuous exposure to stressors could potentially lead to depression [4]. Pharmaceutical intervention aiming toward relevant therapeutic strategies is a field of intense research efforts.

Mitochondria are implicated in anxiety and stress responses [8,9,10,11,12,13,14], with prominent alterations in oxidative phosphorylation and oxidative stress pathways being reported in the literature [15,16,17]. Psychiatric drugs exert their beneficial effects through modulating mitochondria [18,19], whereas it has been previously demonstrated that selective mitochondrial targeting exerts anxiolytic effects in vivo in high-anxiety mice [20]. The necessity for inhibiting the reactive oxygen species (ROS) generated in the mitochondria is eminent, highlighting the necessity to devise novel mitochondria-targeted therapies. A number of drugs that penetrate the mitochondrial membrane and accumulate in the mitochondria, such as Q10 [21], idebenone [22], AICAR [23], and febuxostat [24], have been discovered. Nevertheless, the need for explicitly targeting mitochondria is emerging.

A plethora of mitochondria-targeting molecules have been discovered, with triphenylphosphonium-based compounds (TPPs) being the most efficient [25,26]. Compounds containing the triphenylphosphonium (TPP) moiety have gained significant attention due to their ability to selectively accumulate in the mitochondria, a rapidly emerging approach for sub-cellular drug targeting. TPP conjugates have been studied intensively in alleviating a plethora of diseases, such as cardiovascular malfunction, neurological disorders, and cancer, and influencing mitochondrial metabolism in inflammatory and metabolic diseases, particularly when conjugated with antioxidants such as ubiquinone or plastoquinone. An array of TPP-based drugs have been developed, with MitoQ being the most successful molecule of that category. MitoQ exerts a beneficial effect on brain function [27]. The main mode of MitoQ action on the mitochondria is its high antioxidant activity, which mainly protects from mitochondrial DNA damage [28].

Hydroxytyrosol (HT) is a well-known, naturally occurring, non-toxic polyphenol stemming from the olive tree’s main bioactive component, Oleuropein [29]. HT is one of the most potent natural product-based antioxidants [29,30,31]. A new antioxidant molecule, hereafter named TPP-HT, was synthesized, inspired by HT’s strong activity.

This new antioxidant was designed for enhanced capability of penetrating the mitochondrial membrane, and it is composed of TPP, a six-carbon alkyl spacer, and HT [32]. The spacer allows HT to enter the mitochondrial membrane and exert its antioxidant capacity into the mitochondrial matrix.

The aim of this research was to estimate the concentration of TPP-HT passing through the blood–brain barrier (BBB) and accumulating in the brain, which is an essential property for any compound with potential neurological activity. This is a fundamental aspect that should be investigated before further pharmacological investigations can be pursued, especially in the case of potentially neuroprotective substances. It is noteworthy that TPP-HT has already shown promising pharmacological activity in hyperlipidemic mice [32]. Therefore, a new validated method for the assessment of TPP-HT was developed. Furthermore, it has been shown that MitoQ exhibits hepatoprotective activity, which is crucial in mental health [33,34]. Therefore, an additional method for the evaluation of the content of TPP-HT in the liver was also developed.

## 2. Results

A workflow of the methodology is illustrated in Figure 1, including the experimental part as well as the chemometric methodology.

### 2.1. Instrumental Conditions

A series of optimization procedures regarding the mass spectrometric and chromatographic conditions were undertaken to achieve high sensitivity for the determination of TPP-HT.

#### 2.1.1. Optimized Conditions of Mass Spectrometric Analysis

The optimum ESI conditions acquired were spray voltage 3500 V; sheath gas 25 a.u.; auxiliary gas 10 a.u.; capillary temperature 320 °C; and probe position C, 0.55 mm, affording the highest precursor signal for TPP-HT and TPP simultaneously. The MS/MS spectrum derived from the fragmentation of the precursor ion of TPP-HT is presented in Appendix A, alongside the proposed structures of the product ions generated from the CID procedure. In addition, the proposed MS/MS fragmentation mechanism is illustrated in Appendix A.

#### 2.1.2. Selection of the Chromatographic Mode and the Mobile Phase Requirements

In the instance of overlapping pKa values, the selection of an HILIC column would be necessary. However, the calculated in silico pKa values lead to the selection of an RP column as no overlapping values were observed. The pKa values for the microspecies (alongside the relevant structures) at various pH levels are presented in Appendix A, whereas the corresponding numeric data can be found in Appendix A, both in the Appendix A. The data indicate that 99.3% of the substance exists in a positively ionized state up to pH 7.4, which suggests that the two phenolic groups are protonated. Therefore, the substance exists in a single microspecies form and can be chromatographed as a single peak. Moreover, the logD plot demonstrates that the distribution in the organic phase is nearly constant, with a value of 7.83 up to pH 7.5. The corresponding logD = f(pH) plot is presented in Appendix A. The obtained logD values of the most abundant microspecies (structure 1) for the different pH values are illustrated in Appendix A, indicating adequate retention for pH 7.5, necessitating the selection of a reversed-phase (RP) chromatographic column.

The calculated pKa for low pH values indicates that the molecule is in the non-ionized form at this range and exhibits a high LogD value. Therefore, the compound can be disturbed at the stationary phase, supporting the decision that it can be chromatographed on a C18 column. Additionally, based on the calculated pKa value of TPP-HT, the mobile phase, as well as the extraction solvent, should be acidic. Therefore, the analysis was conducted using a diluted acidic solution and an RP column, yielding satisfactory results. Moreover, the logP value was 7.45, and a corresponding mesh diagram of TPP-HT using Marvin Sketch is presented in Appendix A. The logP increment is mapped on Van der Waals spheres. The dependence of the distribution of the analytes between the mobile phase and the stationary phase and the physicochemical characteristics of the molecule (pka, logP, and logD) are illustrated in Appendix A, which are provided in the Appendix A.

#### 2.1.3. Selection of the Chromatographic Column

By comparing the results obtained for the two reversed-phase (RP) columns in terms of peak shape and intensity, the Acquity BEH column was selected. Specifically, a peak width reduction of more than 30% was observed when the Acquity BEH column was utilized, facilitating the increase in sensitivity by a factor of 1.5. It is important to note that TPP was used as the IS to acquire comparative results between different days. Furthermore, the retention time for TPP-HT was 11.9 min and the capacity factor was calculated and found to be 38.6, whereas the corresponding values for IS were 10.3 min and 32.9, respectively. The selectivity factor was 1.2.

### 2.2. Chemometrics

A gap in the treatment of the data according to the regulatory bodies was realized, taking into consideration that multiple fortified calibration curves have to be employed to evaluate the quantification of the results. This issue concerned the construction of multiple curves that should be represented from a single one to proceed to the quantification.

Initially, the six fortified calibration curves were constructed (Figure 2a). The silhouette coefficient, which is a statistical method that calculates both the intra-cluster and inter-cluster distances to assess the quality of the clusters, and the Dunn index, which is utilized for the identification of groups of clusters that exhibit both compactness and low variance among their members [35], were employed. A comprehensive examination was conducted on four distinct options for cluster sizes, namely 2, 3, 4, and 5. The values obtained for the Dunn Index were 6.1433, 1.5096, 1.5522, and 2.1905, respectively. Furthermore, Silhouette analysis was conducted, yielding values of 0.7691, 0.5871, 0.3799, and 0.2050 for cluster sizes 2, 3, 4, and 5, respectively. These analyses indicated that the cluster size 2 was the optimal choice, as it yielded the highest values with both methods.

Additionally, visual inspection showed that the curves formed two clusters. The equation of these curves was calculated, and the equations (slopes and intercepts, two columns, one for the slope and one for the intercept) as well as their slope and intercept, were calculated separately to examine the clustering among the six fortification curves by cluster analysis employing visualization via dendrograms. The number of clusters was set at 2 based on visual inspection of our data as well as the Dunn and Silhouette metrics calculated. The Canberra distance was chosen as the clustering distance metric, and the complete linkage was used as the agglomeration method. As shown in the corresponding dendrograms (Figure 2b–d), there is a clear distinction between the two days. Consequently, the data coded as 1st, 2nd, and 3rd are assigned to the first cluster (Cluster A—1st day 1st, 2nd, and 3rd replicate), whereas the data coded as 4th, 5th, and 6th are attributed to the second cluster (Cluster B—2nd day 1st, 2nd, and 3rd replicate). It is important to clarify that each curve was constructed individually by spiking TPP-HT at different concentration levels as indicated in the corresponding graph.

Following the visual assessment of discrimination described above, the calibration curves of each day were compared using the extra sum of squares F test. This test allowed for the investigation of the fits of two nested models, with least squares regression. The obtained *p*-values for these two clusters demonstrated that the 1st–3rd could be assigned to Cluster A with a *p*-value of 0.14 and that the 4th–6th could be allocated to Cluster B with a *p*-value of 0.18. (*p*-values < 0.05 indicate statistically significant difference between the curves).

The equivalence of the two methods was then assessed by comparing them using the Deming and the Passing-Bablok regression models. The equation using Deming regression was y = (75.65 ± 2.9) × 10^−2^ x + (4.35 ± 0.77) × 10^−2^ with confidence levels (69 × 10^−2^–83 × 10^−2^) for the slope and ((2.5–6.2) × 10^−2^) for the intercept. The equation using Passing–Bablok regression was y = (76.94 ± 0.64) × 10^−2^ x − (4.30 ± 4.7) × 10^−2^ confidence levels (69 ×10^−2^–81 × 10^−2^) for the slope and ((3.6–5.6) × 10^−2^) for the intercept.

The curves of the comparison between the two methods coded as reference (equation from cluster A) and test method (equation from cluster B) are shown in Figure 3 using Deming (Figure 3a) and Passing–Bablok regression (Figure 3b). Additionally, the Bland–Altman plot, demonstrating the contribution of each point (response) of the curves to the differentiation of the two methods, is shown in Figure 3c.

This leads to the conclusion that a curve should be constructed every day, regardless of the presence of the IS. The back-calculated error for the 1st day fortified curve was assessed using the consensus curve of the 1st day, obtaining a % relative error ranging from 1.26 to 16.86%. Accordingly, the % relative error derived for the back-calculation of the 2nd day ranged from 1.59 to 19.47% for the 2nd curve.

On the other hand, when the back-calculation error was estimated using a calibration curve from another day (i.e., the responses of the 2nd day to the equation of the 1st day), the back-calculation error ranged from 0.3 to 50.3%. In both cases, a higher deviation was observed at the lower and upper limits of the curve, while a lower deviation was observed at the centroid of the curve.

This procedure is proposed as an essential workflow in cases where multiple curves are constructed. Therefore, a rational choice should be made to construct a consensus curve that could be used for quantification purposes.

### 2.3. Validation

The validation was performed for the liver and cerebellum. The results for the liver appear in the main text, while the corresponding results for the cerebellum are in Appendix A, unless they are explicitly found in the main text.

#### 2.3.1. Selectivity

The minimum requirement to define a method as selective is that the analytical signal of the analyte of interest should be less than 20% of the signal of the lowest limit of quantitation (LLOQ). Additionally, in the cases that IS is employed, no interference derived from the matrix should be more than 5% of the signal of the IS. In this analysis, no chromatographic peaks were observed in the blank samples for the SRM transitions at the same retention time as the analytes of interest. Thus, the two methods (for the cerebellum and the liver) were found to be specific for the determination of the TPP-HT samples utilizing TPP as an IS.

#### 2.3.2. Linearity

A linear correlation between the analytical signal and the analytical concentration was found for all the analytical assays. In all cases, the equation is of the following form: y = (a ± sa) x+ (b ± sb), where a is the slope of the curve, b is the intercept on the *y*-axis, and sa and sb are the standard errors. According to GraphPad Prism, the confidence intervals are given for both the slope and the intercept at the 95% level of confidence. The standard error (SE) is calculated using the following equation:(1)$$\mathrm{SE}=\frac{\left(\mathrm{upper}\;\mathrm{limit}-\mathrm{lower}\;\mathrm{limit}\right)}{3.92}$$

Accordingly, the equations of the fortified curves, along with the coefficient of determination (R^2^) for the liver, are presented in Appendix A. The corresponding equations and R^2^ values for the cerebellum are provided in Appendix A. The consensus fortification curves of the 1st day and 2nd for the liver samples are described by the equation y = −(24.96 ± 0.43) × 10^−3^x − (3.29 ± 0.53) × 10^−2^ and y = −(18.81 ± 40.9) ×10^−3^x − (1.38 ± 0.48) ×10^−2^, respectively. Additionally, the consensus fortification curve for the cerebellum samples is described by the equation y = (10.01 ± 0.12) × 10^−2^x − (0.82 ± 0.42) × 10^−2^. According to the ANOVA analysis, the intercept includes the value 0 for the 95% confidence level. Therefore, all the curves are likely to pass through the origin of the axes (0.0). In addition, the coefficient of determination (R^2^) is higher than 0.99, which indicates adequate linearity. Furthermore, the QQ plots for the liver samples, which are presented in Figure 4a,c, show that the residuals are normally distributed. Furthermore, there is no evidence of heteroscedasticity as can be demonstrated by their corresponding residual plots, depicted in Figure 4b,d. Additionally, the QQ plot and the homoscedasticity plot for the cerebellum-fortified curve are presented in Appendix A. The same workflow was followed for both matrices, and the same observations were derived from the fortified curves.

#### 2.3.3. Accuracy (Precision and Trueness)

The accuracy of the method has been evaluated in terms of precision (repeatability and intermediate precision) and trueness for the liver and cerebellum samples. The results for the liver are referred to in the main text (Table 1), while the corresponding ones for the cerebellum are presented in the Appendix A. The average value and the % RSD in terms of repeatability and intermediate precision for 5 fortification levels (2.5-7.5-10-17.5-20 ng g^−1^) corresponding to LLOQ- LQC–MQC–HQC–ULOQ are demonstrated in Table 1 for the evaluation of the repeatability and intermediate precision. The % Rec and the associated confidence interval (95%) are also tabulated in Table 1 for the assessment of trueness. The table shows that the obtained values are in accordance with the requirements of ICHQ2R2 [36] and the FDA bioanalytical methods [37].

The results for both tissues exhibit that the methodology meets the requirements to be utilized for the measurement of TPP-HT.

#### 2.3.4. Matrix Effect

The matrix effect is mainly related to analyses performed with mass spectrometer detectors, and its contribution is significant in the case of complex matrices such as biological materials. The signal of the analyte of interest, TPP -HT, is likely to be affected by the constituents of the sample being analyzed. This phenomenon can affect the ionization, causing either suppression or enhancement of the signal. When the % matrix effect is positive, it indicates ion enhancement; when it is negative, it indicates ion suppression. If the %ME factor is greater than ± 20%, the matrix effect is strong. Τhe % matrix effect calculated for the liver was 14.6, while the corresponding one for the cerebellum was 16.8%.

#### 2.3.5. Carryover

The results of the carryover study showed that less than the signal of the LOD value was observed, showing that no carryover occurs in both matrices.

#### 2.3.6. Limit of Detection and Quantification

The instrumental LOD and LOQ were calculated as 0.9 ng mL^−1^ and 2.9 ng mL^−1^, respectively. In addition, the LOD and LOQ of the method were found to be 1.1 ng g^−1^ and 3.2 ng g^−1^ for liver samples and 0.9 ng g^−1^ and 2.8 ng g^−1^ for cerebellum samples, respectively. Τhe abovementioned methods achieve low limits of detection and quantification, which are appropriate for the purpose of this analysis.

### 2.4. Results of Analysis in Cerebellum and Liver Samples

The presence of TPP-HT in the cerebellum and liver was evaluated by assessing the levels in TPP-HT-treated and non-treated animals. Interestingly, the substance under investigation penetrated the BBB, as a level of (11.45 ± 0.65) ng g^−1^ was found in the cerebellum. In addition, the level of the substance in the liver was found to be (4.82 ± 0.62) ng g^−1^. The results are expressed as (average ± SEM) Taking into consideration that the daily administration dosage, which is 20 mg kg^−1^ day-1, and the average weight of cerebellum and liver being 90 and 1100 mg, respectively, it is calculated that a small percentage (approximately 5%) of TPP-HT passes through the barrier and reaches the cerebellum; a similar percentage (approximately 0.25%) is also transported to the liver. Nevertheless, given that these compounds are targeted to mitochondria, even if found in small quantities, they may accumulate in these organelles, resulting in significant concentration. It is possible that an alternative method of compound administration or other pharmaceutical forms (i.e., suspensions in drinking water) may prove more efficacious in obtaining high levels of compounds to the target organs. It must be noted that a BBB penetration of 1–5% is adequate for a successful drug targeting the brain [38]. Furthermore, a concentration enrichment of 50- to 70-fold is achieved in the mitochondrial matrix due to the selectivity of TPP-based substances to penetrate the extracellular membrane of the target organelle [25].

## 3. Discussion

The objective of the study was to ascertain whether TPP-HT could permeate the BBB. Furthermore, the accumulation in the liver has been examined, as it has been documented that drugs designed to target mitochondria frequently accumulate in the liver. TPP-HT was administered orally to mice through drinking water, and it was of interest to investigate the quantity that successfully traverses the gastric barrier and ultimately reaches the cerebellum and the liver. The latter has been investigated, as recent research has linked hepatic disorders to mental health [39,40,41]. Additionally, MitoQ has been shown to ameliorate such hepatic dysfunction through its mitochondrial healing activity [33,42]. Therefore, it was intriguing to assess the TPP-HT levels in the liver, as the latter shares a similar mode of action to MitoQ, which has been previously utilized in the treatment of glaucoma, thereby establishing a pharmacological efficacy [32].

### 3.1. Selection of Experimental Conditions

#### 3.1.1. Selection of the Detection Methodology

The methodology was developed for two matrices, namely the cerebellum and the liver. Initially, there was no prior knowledge regarding the capacity of TPP-HT to cross through the BBB and reach the cerebellum, as well as whether TPP-HT can accumulate in the liver. Accordingly, it was essential to develop a highly sensitive methodology to achieve the required sensitivity, attaining the lowest possible content levels. The methodology is suitable, even in cases where TPP-HT levels would have been higher, necessitating only a dilution study. Therefore, a method focused on sensitivity has been developed with the purpose of determining low concentrations of the compound, exploring its distribution in the target organelles. It is deemed that a liquid chromatography-tandem mass spectrometry (LC-MS/MS) instrument equipped with a triple quadrupole detector is an appropriate instrument for achieving the appropriate detection and quantification limits.

#### 3.1.2. Selection of the Chromatographic Mode and the Mobile Phase Requirements

To the best of our knowledge, there are no references in the literature for the analytical determination of TPP-HT. An initial investigation was conducted to decide whether a hydrophilic interaction liquid chromatographic column (HILIC) or a reversed-phase column (RP) could be employed for the analysis of this compound.

The molecular structure of the compound contains a fixed positive charge on phosphorus at the alkyl triphenyl phosphine moiety, a lipophilic six-carbon spacer, and two deprotonation sites on the two phenolic groups. This indicates that the molecule could exist in a zwitterionic form, implying a complex pH retention profile. A series of calculations were conducted to gain insight into the nature of the packing material and the pH of the mobile phases.

#### 3.1.3. Optimization of Pretreatment in Liver Samples

For the optimization of the sample preparation procedure, liver tissue has been employed, as its quantity was higher than the available one of the cerebellum. Therefore, a larger number of optimization experiments have been conducted for the first tissue. Among the extraction solvents investigated, better recoveries were obtained when the mixture of acidified ACN and H_2_O was used. Based on the results obtained, higher recovery was obtained for the extraction solvent of ACN: H_2_O-1% ascorbic acid in a ratio of 50:50 (*v*/*v*). Finally, the antioxidant effectiveness was examined by testing ascorbic acid versus citric acid, keeping the ACN: H_2_O ratio at 50:50, (*v*/*v*), showing that ascorbic acid yields better results. Two sample preparation workflows were generated, considering the optimal values obtained. The order in which the experiments were carried out (a-c) and the results obtained are shown in the bar graph in Figure 5. The results are expressed as the ratio of the peak area of the analyte to the peak area of the IS.

Another aim of the study was to develop a common procedure for both tissues. In the course of the study, it has been observed that substituting acetonitrile (ACN) for methanol (MeOH) afforded cleaner extracts for the cerebellum. Thus, the only difference between the two developed pretreatment procedures was the extraction organic solvent, whereas the quantities and the corresponding extraction volumes were adjusted to the available cerebellum quantity. A procedure blank, a liver sample, and the preceding sample fortified at the appropriate level were utilized for each experiment. To ensure comparability of the results, the IS was added to all samples. The sample preparation should be conducted on the same laboratory day as the instrumental analysis because both substances (TPP, TPP-HT) undergo oxidation. No chromatographic peaks corresponding to TPP-HT were observed neither in the procedure blank nor in the sample investigated.

Based on this, two sample preparation workflows were generated. The optimized liver sample preparation is described in the main text (Section 4.5.3), while the cerebellum sample preparation protocol is described in detail in the Appendix A.

### 3.2. Chemometrics

A methodology was developed in accordance with the guidelines set outlined in ICHQ2R1 (bioanalytical method validation and draft EMA) [36,37], and the performance characteristics of the method were evaluated. Furthermore, the construction of multiple curves throughout the analysis is often recommended. In this direction, three fortification curves were constructed on the first laboratory day, employing the optimized sample preparation methodology for liver samples. The same experiment was conducted on the second laboratory day. It was observed that the between-days curves differed despite the use of an IS in both analyses. To obtain more reliable data, a more generic workflow is proposed for the evaluation of the data derived.

To ascertain the equivalence of the six fortification curves and to determine which curve should be employed for quantification purposes, it is necessary to conduct a comparative analysis. The existing methodology (i.e., Bland–Altman) facilitates the comparison of the equivalence of two methods. In the present case, a comparative analysis is to be carried out on six different curves. When a plot is constructed from the six curves generated via Prism, two groups are discerned through an optical evaluation. Prism and sum extended F-squares were utilized, and based on the resulting outcome (*p*-value < 0.05), it can be concluded that the curves are not equivalent. Therefore, they cannot be described by a consensus curve.

The next stage of the investigation was to ascertain whether a trend could be identified and, if so, whether it could be clustered together. This has been investigated utilizing an independent methodology. Due to the existence of a low-dimensional multivariate space, a hierarchical analysis was conducted. Two distinct clusters were observed, leading to the conclusion that curves should be utilized in triplicate (i.e., the first triplicate of the 1st day and the second triplicate of the 2nd day). The subsequent question to be answered was that of the equivalence of each triplicate to utilize a consensus curve. Two extra sum of F-square tests (one test for each triplicate) were conducted to demonstrate the equivalence of each triplicate, thereby enabling their description by a consensus curve. This consensus curve could be used for quantification purposes. The three fortified curves of each cluster, as well as the corresponding cumulative curve for each triplicate, are illustrated in Appendix A. Finally, to validate our findings, additional statistical analysis was conducted on the consensus curves from the two laboratory days, employing Deming regression, Passing–Bablok, and Bland–Altman methods to elucidate discrepancies between the two curves. The difference in slope between the two curves indicates that the first curve underestimates the result, while the second one overestimates it. The back calculation error has been estimated, demonstrating that there is a significant deviation in the low and high points, reaching up to 50%. The proposed workflow entails the construction of multiple curves and the performance of several statistical tests to ensure their equivalence and to obtain reliable results.

The proposed chemometrics workflow followed for the selection of an adequate calibration curve is illustrated in Figure 1.

## 4. Materials and Methods

### 4.1. Compliance with Ethics Guidelines

Male CD1 mice were housed under standard conditions at the animal facility of the University of Ioannina, Greece. Mouse experiments were carried out in accordance with the European Communities Council Directives 2010/63/EU and approved by the local authorities.

### 4.2. In Vivo TPP-HT Administration and Tissue Collection

CD1 mice (*n* = 3) (5–8 weeks old) received TPP-HT orally through drinking water (150 mg L^−1^ diluted in 1.6% aq. EtOH (*v*/*v*)) for 3 weeks. It should be noted that the TPP-HT solution was not administered to control mice (*n* = 3). Upon completion of the treatment, the cerebellum and liver were collected according to standard tissue sampling procedures as previously described [18].

### 4.3. Reagents and Materials

All standards and reagents used were of analytical-grade purity (<95%) unless otherwise specified. MeOH and ACN (LC-MS grade) were purchased from Merck (Darmstadt, Germany), whereas formic acid (FA) 99% was acquired from Fluka (Buchs, Switzerland). Triphenylphosphine hydrochloride (ΤPP), ascorbic acid. and citric acid were bought from Sigma Aldrich (Steinheim, Germany), whereas TPP-HT was synthesized and provided to the Laboratory of Analytical Chemistry by the Department of Pharmacy of National and Kapodistrian University of Athens (NKUA) [32]. The ultrapure water (H_2_O) was provided by a Milli-Q device (Millipore Direct-Q UV, Bedford, MA, USA). Stock solutions of the reference standards (1.0 mg mL^−1^) were prepared in MeOH (LC-MS grade) and stored at −20 °C in ambient glass containers. Working solutions of TPP-HT in various concentrations were prepared by appropriate dilutions of the stock solution with MeOH-acidified with 0.01% FA. Standard calibration curves and tissue-fortified calibration curves, ranging from 1.0 ng mL^−1^ to 50.0 ng mL^−1^, were constructed. TPP was used as the internal standard (IS), and its concentration was kept constant at the 20 ng mL^−1^ level in each working solution.

### 4.4. LC-MS/MS Instrumentation

A TSQ Quantum Access triple quadrupole mass spectrometer equipped with an electrospray ionization (ESI) source was connected to an Accela UHPLC system (Thermo Electron Corporation, San Jose, CA, USA) for the determination of TPP and TPP-HT. An Atlantis T3 column (100 × 2.1 mm, 3 μm) and an Acquity BΕH column (100 mm × 2.1 mm, 1.7 μm) from Waters (Milford, MA, USA), equipped with a guard column, were tested for the chromatographic analysis, and the compounds’ ionization was performed in positive mode (+ESI). The column temperature was maintained at 30 °C throughout the analysis, and the injection volume was set at 10 μL. The mobile phase consisted of (A) MeOH and (B) 0.01% FA. The elution program was gradient, starting with 10% solvent A and remaining constant for 1 min. It was then gradually increased to 100% in 9 min and held constant for 3 min. The initial conditions were restored within 0.1 min, and the column was re-equilibrated for 2 min before the next injection. The flow rate was set to 0.1 mL min^−1^, and the total chromatographic time for each injection was 15 min. Thermo Fisher Scientific Xcalibur software, version 2.3 (Waltham, MA, USA), was used for instrument control and data acquisition.

The parent ion of the analytes and their corresponding products, collision energy (CE), and tube lens (TL) settings were obtained after direct infusion of 1 mg L^−1^ of individual standard solutions in MeOH acidified with 0.01% FA. Selected reaction monitoring (SRM) was used, and two transitions were selected for each analyte. The most abundant ion, defined as the quantifier, and the second most abundant ion, defined as the qualifier ion, were used for quantification and confirmation, respectively. The above parameters are tabulated in Table 2. Furthermore, the ESI settings, spray voltage, sheath gas, auxiliary gas, and capillary temperature were operated at 3500 V, 25 a.u., 10 a.u., and 320 °C, respectively. The probe position of the ESI was set to position C (horizontal probe position) and 0.55 mm (vertical probe position).

### 4.5. Method Optimization

The developed method was optimized in terms of mass spectrometry, liquid chromatography, and sample preparation.

#### 4.5.1. Optimization of ESI Source Conditions

The method was optimized in terms of MS parameters by loop injection to obtain maximum signal intensity of the analyte TPP-HT’s precursor ion. The analytical column was initially equilibrated in 100% MeOH, and the flow rate of the mobile phase was set at 100 μL min^−1^. The composition of the mobile phase was chosen considering the polarity of the investigated analyte. In addition, a flow rate of 100 μL min^−1^ was selected based on our experience regarding the optimum performance of the ESI source with respect to the low flow rate.

As it is important to optimize the MS conditions, specific parameters were evaluated to study their effects on the intensity of the precursor ion. The examined values were 2500-3000-3500 V for the spray voltage, 20-25-30-35-40-45 arbitrary units (a.u.) for the sheath gas, and 5-10-15 a.u. for the auxiliary gas. In addition, the capillary temperature, which is a critical factor for the ionization efficiency of the analytes, was tested by increasing the temperature from 270 °C to 320 °C (270-290-320-350 °C). Finally, the vertical and horizontal positions of the ESI probe were investigated using a combination of vertical (0.55-1.0-1.25-1.75-2.0) and horizontal (positions B and C) distances.

#### 4.5.2. Optimization of Liquid Chromatographic Conditions

##### In Silico Calculations

A calculation was carried out to determine the protonation (pKa) and the lipophilicity (logD) of TPP-HT versus pH using the MarvinSketch software (version 24.3.0). “Marvin was used for drawing, displaying and characterizing chemical structures, substructures and reactions, Marvin 24.3.0, Chemaxon (https://www.chemaxon.com, accessed on 7 October 2024) [43]. The calculation of the pKa values was based on the macro mode 298 K microspecies algorithm, whereas for the calculation of logD, the ChemAxon methodology was employed.

##### Selection of Chromatographic Column

Initially, an RP column, namely Atlantis T3 (3 μm, 100 mm × 2.1 mm), was selected, and a general elution protocol for semipolar compounds was compiled. The LC gradient elution program is described in detail in Section 4.4 as affording adequate retention of the analyte to the chromatographic column. In order to optimize the chromatographic characteristics of the method, 3 different mobile phase compositions were tested, namely (a) (A) MeOH-(B) FA 0.01% (*v*/*v*), (b) (A) acidified MeOH (0.1% FA)-(B) FA 0.01% (*v*/*v*), and (c) (A) ACN, (B) FA 0.01% (*v*/*v*). The best chromatographic characteristics were obtained for the mobile phase consisting of (a). Considering the above result, another RP column with a smaller particle size, specifically the Acquity BΕH column (100 mm × 2.1 mm, 1.7 μm) from Waters (Milford, MA, USA), was studied.

#### 4.5.3. Sample Preparation

##### Optimization of Pretreatment in Liver Samples

Initially, the selection of the organic solvent (MeOH or ACN) was evaluated. Therefore, organic solvents mixtures were tested, i.e., (a) acidified MeOH with 0.01% FA (*v*/*v*), (b) acidified ACN with 0.01% FA (*v*/*v*), and (c) (H_2_O-0.01% FA (*v*/*v*): acidified CAN-0.01% FA (*v*/*v*), 50:50 (*v*/*v*)). Considering the lipophilicity of TPP-HT, the effect of the water percentage was studied by testing five combinations, namely the percentages of H_2_O-ACN of 0-100, 25-75, 50-50, 75-25, and 100-0 were assessed. It is important to note that in the latter experiment, all solutions contained 1% ascorbic acid to inhibit the oxidation procedure.

The optimized liver sample preparation is described in the main text, while the cerebellum sample preparation protocol is described in detail in the Appendix A.

##### Optimized Liver Sample Preparation

In a 2 mL Eppendorf tube, 30 mg of liver tissue was weighed, and 2 μL TPP (1 ng μL^−1^), which was used as the IS, was added to each sample. For quantification purposes, spiking (by adding 3 μL of 0.33 ng μL^−1^ TPP-HT) was performed. The samples were left at room temperature for 1 h after spiking with the analyte and its corresponding IS to facilitate their absorption into the matrix. Afterwards, the addition of 300 μL ACN, 1% asc. acid: H_2_O (50:50, *v*/*v*) acidified with 0.01% FA followed. The samples were agitated on a horizontal shaker for 15 min and centrifuged at 3000 rcf at 4 °C. The supernatants were aspirated into new Eppendorf tubes and stored at −20 °C for 16 h. The samples were then centrifuged and the supernatants evaporated to dryness under a gentle nitrogen stream at room temperature. Finally, the residues were reconstituted by adding 0.1 mL MeOH acidified with 0.01% FA, and the extracts were transferred to 2 mL autosampler glass vials and injected into the liquid chromatography–mass spectrometry (LC-MS/MS) system.

### 4.6. Chemometrics Methodology

The calibration curves were constructed using the demo version of GraphPad Prism (GraphPad software LLC, Boston, MA, USA, GraphPad Prism version 9.3), and we employed an extra sum of F-square test for comparison, whereas the D’Agostino–Pearson and the Shapiro–Wilk tests were implemented for the detection of homoscedasticity trends exhibited by the outliers. Mephas software (version 1.1) [44] was employed for the purpose of clustering the results of the aforementioned comparison, with hierarchical clustering and the Canberra distance being utilized for this process. Consequently, the calculation of these metrics was performed using the R programming language (version 4.3.3) via the RStudio interface (RStudio 2024.12.1 Build 563) [45,46] and specifically the clValid package [47]. Furthermore, GraphPad Prism software was utilized for the construction of the consensus calibration curves. Finally, the curves were compared with the Passing–Bablock, Deming regression, and Bland–Altman plots. The equivalence of the two calibration curves was then assessed by comparing them using the web application ‘Method Comparison & Bias Estimation using R and Shiny’ (online available on https://bahar.shinyapps.io/method_compare/, accessed on 9 October 2024) [48].

### 4.7. Methods Validation and Acceptance Criteria

Two sample preparations, one for the cerebellum and one for the liver, were developed. These methods were validated through the evaluation of the following analytical parameters: selectivity, linearity, trueness (recovery), precision (repeatability and intermediate precision), limit of detection (LOD) and limit of quantification (LOQ) of the method, matrix effect, carryover, and stability. The ratio of the analyte peak area to the IS peak area was calculated at each corresponding level and was used throughout all the calculations as the dependent variable.

#### 4.7.1. Selectivity

The selectivity of the methods (liver and cerebellum) was tested for plausible interferences by analyzing six blanks of the liver and six blank samples of the cerebellum from different mice using the SRM transitions of TPP-HT and TPP at the analytes’ retention time.

#### 4.7.2. Linearity

The linear range of the analyte of our interest was investigated by constructing 8-point fortified curves. Thus, fortified samples were prepared for both matrices at concentrations corresponding to the following levels: 2.5-5.0-7.5-10.0-12.0-15.0-16.0-20.0 ng g^−1^. For each measurement, the ratio of the analyte peak area (TPP-HT) to the internal standard (IS) peak area (TPP) was calculated. The calculation of the linear model was performed using least squares regression with the (0.0) level excluded, and the calibration curve was not forced to pass through it. The linear range of the curves was investigated over two laboratory days in triplicate for the liver samples. Based on the aforementioned results, and considering the limited cerebellum quantity as well as the acquired knowledge from the liver experiment, the linearity was assessed in triplicate during a laboratory day for cerebellum samples.

#### 4.7.3. Accuracy (Trueness and Precision)

The accuracy of the methods was evaluated by assessing trueness as a percentage of recovery (%Rec) and precision as repeatability and intermediate precision, expressed as a percentage of relative standard deviation (%RSD). The methods’ accuracy was evaluated at 5 different concentration levels via the analysis of fortified samples at 5 levels per substrate (cerebellum and liver). Namely, the levels were 2.5 ng g^−1^ (lower limit of quantification, LLOQ), 7.5 ng g^−1^ (lower quality control, LQC), 10 ng g^−1^ (middle quality control, MQC), 16 ng g^−1^ (higher quality control, HQC), and 20 ng g^−1^ (upper limit of quantification, ULOQ). Six replicates of each fortification level were conducted for the estimation of repeatability and nine replicates for the estimation of intermediate precision. The % Rec was calculated based on the aforementioned experiment. For each analytical determination and at each concentration level, the mean of the experimental data was calculated, as well as the %RSD, the % Rec, and its associated interval at the confidence level of 95%.

#### 4.7.4. Matrix Effect

The matrix effect was evaluated through the calculation of the ratio of the reference standard curve’s slope divided by the corresponding calibration curve constructed by post-spiking blank liver or cerebellum samples.

#### 4.7.5. Carryover

The carryover test was conducted by means of sequential injection of blank liver and cerebellum samples after the injection of a fortified liver sample at the level of ULOQ.

#### 4.7.6. Limit of Detection and Quantification

The instrumental limits of detection and quantification were calculated from the calibration curve of the reference standards. The LOD and LOQ values were calculated based on the following equations:(2)$$\mathrm{LOD}=\frac{3.3\times \;\mathrm{SD}}{\mathrm{slope}}$$(3)$$\mathrm{LOQ}=\frac{10\times \;\mathrm{SD}}{\mathrm{slope}}$$
where SD: standard error of the intercept.

In addition, the method LOD and LOQ were calculated based on the signal-to-noise ratio (S/N) of the fortified curves.

## 5. Conclusions

A new antioxidant, TPP-HT, was synthesized and orally administered to mice through drinking water. Two LC-MS/MS methods were developed and optimized through mass spectrometry, liquid chromatography, and sample preparation for the determination of TPP-HT, one in the mouse cerebellum and one in the mouse liver. A chemometric workflow for comparing calibration curves revealed the necessity of the construction a new curve daily, irrespective of the presence of the IS, due to oxidation of the compounds. This approach proposes a way to systemically compare calibration curves and unify those with high statistical similarity to one consensus curve, selecting the most appropriate one. The methods were validated, meeting the required criteria of selectivity, linearity, accuracy, carryover, and matrix effect. Furthermore, TPP-HT was found to penetrate the BBB, with a level of (11.45 ± 0.65) ng g^−1^ being detected in the cerebellum. In addition, TPP-HT was also found in mouse liver at a level of (4.82 ± 0.62) ng g^−1^. Thus, it is demonstrated that TPP-HT accumulates in the cerebellum at a rate that is more than two-fold higher than that observed in the liver and can therefore be used in vivo experiments to modulate brain mitochondria in neuropsychiatric phenotypes.

## Figures and Tables

**Figure 1 molecules-30-01900-f001:**
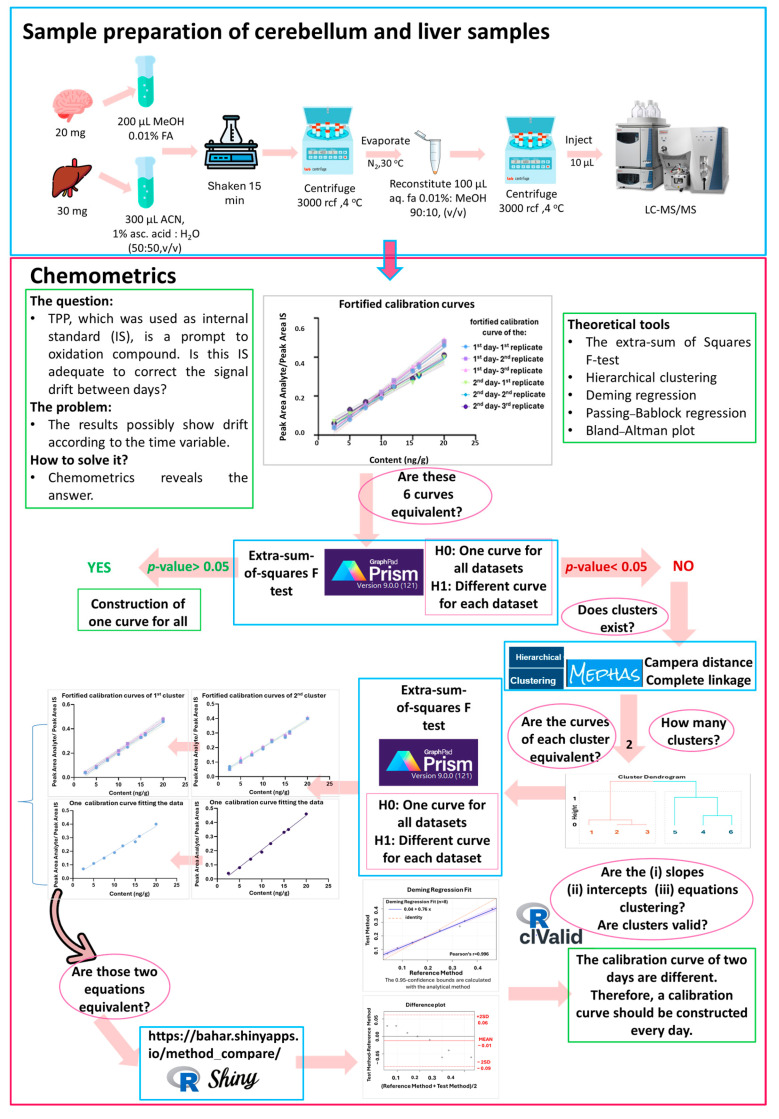
Innovative chemometric workflow for the assessment of representative calibration curve.

**Figure 2 molecules-30-01900-f002:**
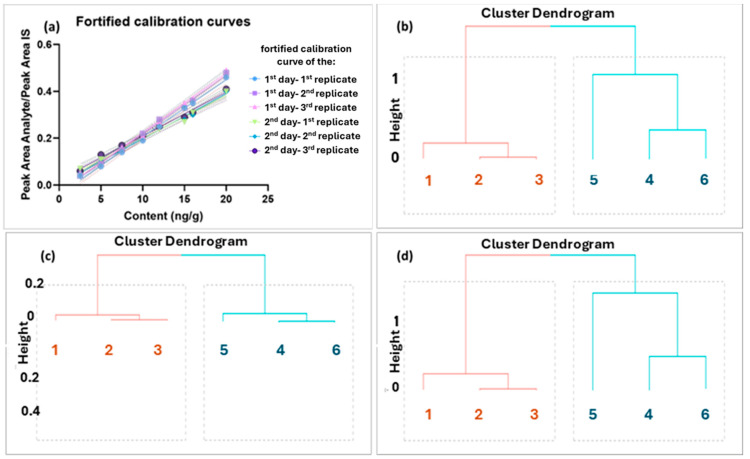
(**a**) Fortified calibration curves for 2 days, 3 replicates for each day, (**b**) dendrogram for equations’ clustering analysis (slopes and intercept as variables), (**c**) dendrogram for slopes’ clustering analysis, and (**d**) dendrogram for intercepts’ clustering analysis of the fortified calibration curves using Canberra distance. Two groups are derived from each dendrogram highlighted in orange and light blue, respectively (each one belonging to the analysis day), and each group was used to create one daily consensus curve.

**Figure 3 molecules-30-01900-f003:**
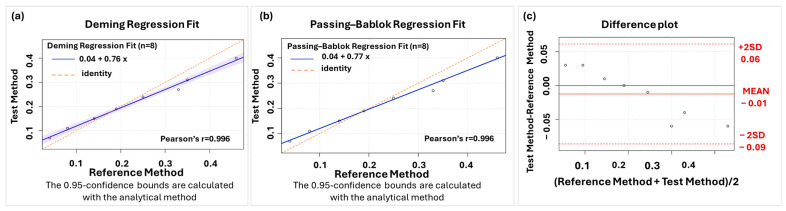
(**a**) Deming regression fit, (**b**) Passing–Bablok regression fit, (**c**) Bland–Altman plot. The slope indicates a proportional difference between the two methods.

**Figure 4 molecules-30-01900-f004:**
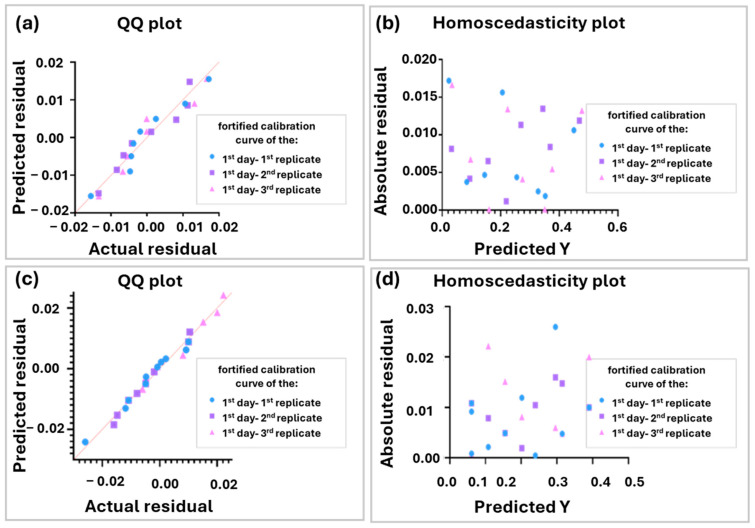
(**a**) QQ plot of day 1 fortified curves. (**b**) Homoscedasticity plot of the day 1 fortified curves. (**c**) QQ plot of day 2 fortified curves. (**d**) Homoscedasticity plot of day 2 fortified curves for liver samples. The QQ plots show that the residuals are normally distributed, and the homoscedasticity plots demonstrate the homoscedasticity of the residuals.

**Figure 5 molecules-30-01900-f005:**
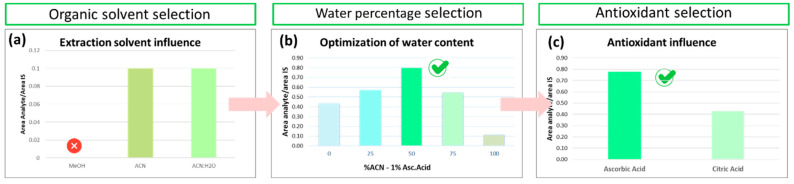
Bar graphs for the comparison of (**a**) organic extraction solvent, where the ⊗ symbol in MeOH declares the absence of analytical signal, when this solvent has tested (**b**) water percentage, where the ✓ symbol in circle highlights the optimum one and (**c**) antioxidant selection where the ✓ symbol in circle highlights the optimum one.

**Table 1 molecules-30-01900-t001:** Results of accuracy (precision and trueness) in 5 fortification levels for liver samples.

	Repeatability (*n* = 6)		Intermediate Precision (*n* = 9)		Trueness (*n* = 6)
Fortification Level (ng g^−1^)	Average (ng g^−1^)	%RSD	Average(ng g^−1^)	%RSD	%Recovery (Average ± Confidence Interval (95%))
2.5 (LLOQ)	2.63	4.4	2.67	5.7	106 ± 9.0
7.5 (LQC)	7.35	5.5	7.21	9.7	97 ± 11
10 (MQC)	9.34	5.0	9.19	6.1	93 ± 9.1
17.5 (HQC)	16.61	3.7	16.55	3.3	104 ± 7.5
20 (ULOQ)	20.06	3.1	20.10	2.8	100 ± 6.2

**Table 2 molecules-30-01900-t002:** Selected reaction monitoring (SRM) transitions οf TPP and TPP-HT.

Compound	Precursor Ion (*m*/*z*)	Quantifier Ion (*m*/*z*)	CE (eV)	Qualifier Ion (*m*/*z*)	CE (eV)	Tube Lens
TPP-HT	499.2	182.7	67	261.4	42	126
TPP	307.0	182.8	42	184.8	24	78

## Data Availability

Data are available upon request.

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
