# Peer review of "Liquid Chromatography-Tandem Mass Spectrometry Method Development and Validation for the Determination of a New Mitochondrial Antioxidant in Mouse Liver and Cerebellum, Employing Advanced Chemometrics"

_molecules, 2025, doi:10.3390/molecules30091900_

Round 1
Reviewer 1 Report
Comments and Suggestions for Authors
This manuscript presents a method for evaluating the concentration of TPP-HT crossing the blood-brain barrier (BBB) through experimental and chemometric approaches. The study optimizes both the experimental process and the chemometric methods. The manuscript provides a well-detailed and reasonable methodology. Before publishing, some minor issues should be addressed:
1. Figure 5 or an additional overall flowchart should be placed earlier in the Results section to provide readers with a clearer understanding of the overall workflow.
2. This flowchart should also include both the experimental and chemometric aspects to present a comprehensive view of the method.
3. A general descriptive summary should accompany the flowchart to explain the research workflow concisely.
4. Subjective and unverifiable descriptive terms such as “novel” and “advanced” should be removed or replaced with more objective and substantiated expressions.
5. The font size of the axes in Figures 1 and 2 should be increased for better readability.
6. The resolution of these figures needs to be improved to ensure clarity when enlarged or printed.
Author Response
Dear Reviewer,
Thank you for your insightful comment. Please find attached the responses to your comments.
Kind regards,
Dr. Anthi Panara

Reviewer 2 Report
Comments and Suggestions for Authors
After reviewing the manuscript “ LC-MS/MS method development and validation for the determination of a new mitochondrial antioxidant in mouse liver and cerebellum, employing advanced chemometrics”, I have the following comments:
- The study developed and validated an LC-MS/MS method to quantify a novel antioxidant, TPP-HT, in mouse liver and cerebellum. Results confirmed its successful oral absorption and blood-brain barrier penetration.
- The novelty of TPP-HT needs stronger justification. The claim that this is a “new antioxidant” is somewhat vague without structural characterization, novelty claims, or broader pharmacokinetic/pharmacodynamic context.
- Chemometric workflow: The clustering methodology lacks justification for using two clusters based only on visual inspection and dendrogram distances. A quantitative clustering validation metric (e.g., silhouette score, Davies-Bouldin index) would improve rigor.
- While it’s suggested that a calibration curve must be constructed every day, the impact of IS (TPP) on normalizing inter-day variation is downplayed. If IS is not sufficient to correct day-to-day variability, this undermines the method's robustness.
- The calculation that 1% of TPP-HT reaches the cerebellum and 0.25% the liver is based on the final ng/g measurement but does not account for mouse tissue weights, bioavailability, or metabolism. The bioanalytical concentration-to-percentage conversion lacks transparency.
- Language issues:
The compound is referred to variously as “TPP-HT,” “the molecule,” or “the substance under investigation” without clear transitions.
“We that TPP-HT has already…” (line 90) is grammatically incorrect.
The usage of symbols and units should be consistent throughout the paper.
- Figure 1: The figure could benefit from clearer labels (e.g., “Day 1 – S1, S2, S3”) to directly show which curves belong to which group in part (a). Also, the caption should explicitly mention what each panel represents and explain that these analyses were used to justify grouping calibration curves for consensus curve generation.
- No animal number (n) is reported. This limits reproducibility and ethical transparency.
- Figure S6: The visual confirmation of normality (from the QQ plot) and the constant variance (from the homoscedasticity plot) is assumed, but not critically evaluated in text. Ideally, deviations from the straight line (QQ) or any funnel shape (residuals vs. fitted) should be discussed. Also, the caption lacks interpretive detail. It should explain what statistical assumptions these plots evaluate and how the results support the model’s validity.
- Conclusion section: While the paper involves a rigorously validated LC-MS/MS method, the conclusion does not emphasize the robustness or novelty of the analytical and chemometric workflow, which is one of the most innovative aspects of the work.
Author Response

(The authors gave the same response as above.)

Round 2
Reviewer 2 Report
Comments and Suggestions for Authors
The authors have clearly made a concerted effort to address all reviewer feedback in a thoughtful and detailed manner. The novelty of TPP-HT is now better contextualized, and the statistical validation added to support the chemometric clustering strengthens the analytical approach. The discussion around the role and limitations of the internal standard is appropriately clarified, and the inclusion of animal numbers improves transparency.
Overall, the manuscript is well-structured and scientifically sound. The LC-MS/MS method and chemometric workflow are both robust and relevant. Minor language or formatting issues do not affect the quality of the work and can be addressed during the final editing stage. I recommend the manuscript for acceptance.
Minor Issues:
- Mixed use of "Fa" for formic acid should be standardized to “FA”.
- Some figure references (e.g., S3–S6) are scattered and may benefit from better integration with the narrative in the main text.
- Ensure all supplementary figures are captioned clearly and that they are uploaded correctly during submission.
- The use of Greek symbols and superscripts (e.g., ng g⁻¹, °C) is inconsistent in a few places.
- The chemometric workflow is described in detail in both Results and Discussion, leading to some repetition. A clearer distinction between results and interpretation would help.
Author Response
Please find attached the responses to the second reviewer's comments.
Kind regards,
Dr. Anthi Panara
